# Secreted Non-Coding RNAs: Functional Impact on the Tumor Microenvironment and Clinical Relevance in Triple-Negative Breast Cancer

**DOI:** 10.3390/ncrna8010005

**Published:** 2022-01-11

**Authors:** Silvia Di Agostino, Mahrou Vahabi, Chiara Turco, Giulia Fontemaggi

**Affiliations:** 1Department of Health Sciences, “Magna Graecia” University of Catanzaro, 88100 Catanzaro, Italy; 2Oncogenomic and Epigenetic Unit, IRCCS Regina Elena National Cancer Institute, 00144 Rome, Italy; mahrou.vahabi@gmail.com (M.V.); chiara.turco@ifo.gov.it (C.T.)

**Keywords:** breast cancer, TNBC, non-coding RNA, ncRNA, exosomes, tumor microenvironment, liquid biopsy

## Abstract

Triple-negative breast cancer (TNBC) is a subtype of breast carcinoma characterized by poor prognosis and high rate of metastasis. Current treatment is based on chemo- and/or radiotherapy and surgery. TNBC is devoid of estrogen, progesterone and HER2 receptors. Although precision medicine has come a long way to ameliorate breast cancer disease management, targeted therapies for the treatment of TNBC patients are still limited. Mounting evidence has shown that non-coding RNAs (ncRNAs) drive many oncogenic processes at the basis of increased proliferation, invasion and angiogenesis in TNBC, strongly contributing to tumor progression and resistance to treatments. Many of these ncRNAs are secreted in the tumor microenvironment (TME) and impinge on the activity of the diverse immune and stromal cell types infiltrating the TME. Importantly, secreted ncRNAs may be detected as circulating molecules in serum/plasma from cancer patients and are emerging a promising diagnostic/therapeutic tools in TNBC. This review aims to discuss novel insights about the role of secreted circulating ncRNAs in the intercellular communication in the tumor microenvironment and their potential clinical use as diagnostic and prognostic non-invasive biomarkers in TNBC.

## 1. Introduction

Breast cancer is the second leading cause of cancer-related mortality in women. Triple-negative breast cancer (TNBC) is an aggressive subtype of breast cancer characterized by significant inter- and intra-tumor molecular heterogeneity and defined by the lack of expression of estrogen receptor (ER), progesterone receptor (PR), and HER2 receptor. Owing to the absence of receptors, exploited in other BC subtypes as therapeutic targets, TNBC patients often experience less favorable outcomes compared to other breast cancer subtypes [1,2]. The current standard therapies are surgical and medical treatment followed by adjuvant radiotherapy (RT) with or without chemotherapy. Compared to other subtypes of breast cancer, TNBC has the highest amount of metastasis and lowest survival rates [3]. There is a high risk of recurrence in this subtype of breast cancer also; resistance to therapy is the major obstacle to successful treatment of TNBC [4]. Consequently, there is limited progress in targeted therapies for the treatment of TNBC patients. This indicates the need to develop new TNBC treatment strategies.

Many studies have shown that non-coding RNAs make up the majority (about 90%) of the transcribed genome, and they lack the ability to code for proteins. Surprisingly, only 1–2% of the transcribed genome encodes proteins [5]. Various non-coding RNAs have been shown to regulate gene expression. Non-coding RNAs comprise ribosomal RNAs (rRNAs), transfer RNAs (tRNAs), microRNAs (miRNAs), small interfering RNAs (siRNAs), small nuclear RNAs (snRNAs), small nucleolar RNAs (snoRNAs), PIWI-interacting RNAs (piRNAs), extracellular RNAs (exRNAs), small Cajal body-specific RNAs (scaRNAs), long non-coding RNAs (lncRNA) and circular RNAs (circRNAs). These non-coding RNAs play a crucial role in many biological processes [5,6]. Moreover, their aberrant expression can lead to cancer initiation, progression, and metastasis, therefore they might be considered as therapeutic targets and attractive tools for diagnosis and prognosis of cancer [5,6].

Although high morbidity tumor tissue biopsy is currently the most widely used procedure for cancer diagnosis and definition of prognosis, attempts to find promising non-invasive biomarkers for cancer screening, diagnosis and prognosis are generating considerable interest [7]. Of note, non-coding RNAs can circulate in the body fluids and can be detected easily in the plasma, serum, saliva, seminal, cerebrospinal fluid and urine of cancer patients including triple negative breast cancer [7,8,9]. Importantly, circulating non-coding RNAs represent a source of information about the status of malignancy and changes of circulating ncRNAs levels are correlated with the degree of tumor progression [7,8].

Among circulating non-coding RNAs, microRNAs, long non-coding RNAs and circular RNAs have been considered as a biomarker in different types of cancer including TNBC. microRNAs are the most studied group of non-coding RNAs [9]. These small non-coding RNAs (17–22 nucleotides) act as a gene expression regulator by binding to 3′UTR of target mRNAs and recruiting specific silencing proteins that form the RISC (RNA Induced Silencing Complex) [10]. Many studies have reported that microRNAs are considered not only as regulatory molecules, but also as potential therapeutic targets [11,12,13,14]. Long non-coding RNAs (lncRNAs) are transcripts of RNAs longer than 200 nucleotides, which do not encode proteins. LncRNAs can function as signaling, decoys, guided, and scaffold lncRNAs [15,16]. Circular RNAs (circRNAs) have been recently discovered and are generated by non-canonical back-splicing events, frequently linked to exon-skipping of pre-mRNA. Back-splicing leads to the production of covalently closed circular RNAs which lack 3′ end poly (A) tail and 5′ end cap [17]. Due to their structure, circRNAs are not exposed to the majority of RNases and therefore are very stable molecules. Increasing observations have indicated that they can act as sponges, which control the activity of other regulatory proteins like RNA binding proteins (RBPs) or microRNAs. Moreover, they can function as gene expression and transcription regulators [18,19,20].

Circulating non-coding RNAs can be released into body fluids, and reach distant target organs, as cell-free, non-vesicle associated molecules or otherwise as extracellular vesicle-associated molecules. In the first case, proteins like Argonaute2 (AGO), GW182, nucleophosmin 1 (NPM1) and high-density lipoproteins (HDL) may be responsible for the binding and transport of non-coding RNAs. In the second case, extracellular vesicles (EVs) surrounded by lipid bilayer can transport the non-coding RNAs. EVs can be classified into two large classes: (a) exosomes, which are produced by exocytosis from a multivesicular body (MVB) with an approximate size of 40–100 nm [21]; and (b) microvesicles (MVs), which derive directly from the plasma membrane through outward budding with dimensions in the range of 100–1000 nm [8,22].

Extracellular vesicles containing a variety of macromolecules, included non-coding RNAs, are released in the tumor microenvironment (TME), where they actively participate in cancer progression (Figure 1).

The tumor microenvironment (TME) is a complex environment composed of a variety of cell types including tumor cells, tumor stromal cells like fibroblasts, endothelial and immune cells [23]. In general, there is a crosstalk between cancer cells and the TME, therefore the TME affects the development and progression of cancer [24,25]. Exosomes have been now recognized as major mediators of the communication between tumor cells and the other cell types in the TME. Additionally, they have the ability to modulate the behavior of cells populating the TME and to cooperate in immune response and tumor metastasis [24]. Therefore, exosomes containing non-coding RNAs could be exploited for liquid biopsy and used as promising non-invasive biomarkers for the diagnosis and the definition of prognosis in cancer. In this review, we provide an overview of the recent literature on the role of circulating non-coding RNAs in the cross-talk between TNBC cells and the TME, as well as on the possible use of these ncRNAs as biomarkers and indicators of tumor progression.

## 2. Functional Impact of Secreted ncRNAs on Surrounding Stromal Cells and at Metastatic Sites in TNBC

It is well recognized that cancer cells communicate, directly by cell–cell contact or through paracrine mechanisms, with each other and with the other cells of the TME, causing a gene expression reprogramming that enables protumoral functions, such as neoangiogenesis and immune escape. Extracellular vesicles, in particular, have been shown to exert a pivotal role in the transport of non-coding RNAs from cancer cells to neighboring cancer and stromal cells, contributing to the functional reshaping of the TME [26].

The majority of the studies have focused on the identification of microRNAs secreted by cancer cells and targeting a variety of cell types in the TME. It has been extensively reported that the miRNA profiles of EVs are distinct from the matched cellular profiles. Of note, cancer cells are able to actively secrete high amounts of specific miRNAs that are usually retained in normal untransformed cells [27]. Moreover, metastatic breast cancer cells have been shown to secrete exosomes that are enriched in miRNAs compared to non-metastatic breast cancer cells [28]. EVs isolated from cancer cells or from the blood of cancer patients enclose not only miRNAs, but also protein components of the RISC complex, thus displaying a cell-independent capacity to process pre-miRNAs into mature miRNAs and an immediately active miRNA-RISC complex in the recipient cell [28].

One major function of miRNA-containing EVs secreted by metastatic TNBC cells is to confer metastatic capability to the non-metastatic cancer cell population. It has been reported, for example, that miR-200 is secreted in EVs from TNBC cells and is detectable in the serum of metastatic breast cancer patients [29,30]. Transfer of miR-200 from metastatic to non-metastatic cells impinges on these last by inducing mesenchymal-to-epithelial transition. Mechanistically, miR-200 not only enhances epithelial traits, facilitating engraftment in the metastatic niche, but also suppresses the secretion of anti-metastatic factors in recipient cells [31]. A similar pro-invasive activity of recipient cells has been reported for EVs-associated miR-10b in TNBC [32] (Figure 2).

As already mentioned, non-tumor cell types in the tumor stroma may also be targeted by cancer-derived EVs. A cell type that is strongly reprogrammed in the TME of primary tumors and at metastatic sites is the cancer-associated fibroblast (CAF), which contributes to cancer progression through its ability to affect extracellular matrix composition, T-cell function and growth factors secretion (reviewed in [33]).

A pivotal role in CAF reprogramming is exerted in TNBC by miR-9. The oncoprotein MYC is responsible for the induction of miR-9 in TNBC cells, where miR-9 targets E-cadherin (CDH1) 3’-UTR thus favoring EMT, motility and metastasization [34]. Of note, analysis of CAFs isolated from TNBC patients evidenced that miR-9 is expressed in these cells at higher levels, compared to matched normal fibroblasts. miR-9 is indeed secreted in exosomes by cancer cells and transferred to recipient fibroblasts in the TME, resulting in enhanced cell motility of CAFs. Expression of miR-9 in CAFs causes the modulation of genes mainly involved in cell motility and extracellular matrix remodeling pathways [35]. Further study from the same group also showed that miR-9 directly targets the ECM glycoprotein fibulin-3 (EFEMP1) in CAFs and EFEMP1 down-regulation is responsible for the observed increased CAFs motility upon miR-9 induction. Moreover, the supernatant of EFEMP1-depleted CAFs is able to confer resistance to cisplatin to TNBC cells, highlighting a two-way paracrine communication between these two cell types [36].

Another miRNA able to impact on CAFs behavior in the TME is miR-105. Similarly to miR-9, miR-105 is also induced by the oncoprotein MYC in TNBC cells; this miRNA is subsequently encapsulated in extracellular vesicles and secreted by the tumor cell, then transferred in paracrine manner to CAFs, also causing activation of MYC pathway in these recipient cells. Specifically, CAFs undergo a reprogramming whereby, in the presence of nutrients, they enhance glucose and glutamine metabolism to fuel adjacent cancer cells, while, upon nutrients’ deprivation, these CAFs detoxify metabolic wastes by converting lactic acid and ammonium into energy-rich metabolites [37]. miR-105-dependent reprogramming of CAFs thus influences the composition of the shared metabolic environment to promote cancer cells’ growth. Interestingly, secreted miR-105 not only affects CAFs behavior, but also strongly impacts on endothelial cells. Indeed, in endothelial monolayers, the transfer of cancer-secreted miR-105 causes down-regulation of tight junction protein ZO-1 and a consequent destruction of tight junctions allowing metastasization of breast cancer cells [38]. Permeability of the endothelial monolayer is also enhanced by the EVs-associated miR-939, targeting VE-cadherin [39].

The endothelial function is strongly affected by the EVs of breast cancer cell derivation also at metastatic sites. Breast cancer cells metastasizing to the brain may indeed promote the destruction of the blood–brain barrier (BBB) through the secretion of EV-associated miR-181c, which in turn down-regulates 3-phosphoinositide-dependent protein kinase-1 (PDPK1). PDPK1 reduction then leads to activated cofilin-induced modulation of actin dynamics responsible for the BBB modification facilitating the metastatic process [40].

In the context of metastatic breast cancer, another recipient cell type of cancer-derived EVs is the osteoblast. Bone represents one of the most frequent metastatic sites of advanced breast cancer. Wang and colleagues reported that breast cancer-derived miR-218 is present in the blood of breast cancer patients with bone metastases. Functionally, cancer-secreted miR-218 directly down-regulates type I collagen (COL1A1) expression in osteoblasts, inhibiting its deposition in the bone [41]. Reprogramming of CAF by cancer-derived EVs has been also reported to occur in the lung, another frequently targeted organ in TNBC. Specifically, EV-associated miR-122, secreted by breast cancer cells, is able to reprogram lung fibroblasts and astrocytes to suppress glucose metabolism by modulating the expression of pyruvate kinase (PKM); this network down-regulates glucose consumption in niche cells to allow metastasized cancer cells to have more glucose available [42].

In addition to the above-mentioned miRNAs, lncRNAs have also been identified as secreted in exosomes and functionally relevant in the TME (summarized in Figure 3). One such example is represented by SNHG16, a breast cancer-derived exosomal lncRNA, which may reach the Tregs population in the TME and cause immunosuppression. Functionally, SNHG16 is able to sponge miR-16-5p leading to activation of SMAD5 expression; activated SMAD5, in turn, upregulates the expression of CD73 in Tregs, a feature associated to the immunosuppressive function [43]. Another lncRNA secreted in exosomes from breast cancer cells, included TNBC cells, is BCRT1. BCRT1, transferred in exosomes, reaches the TAM population in the TME, causing their M2 polarization, which in turn accelerates cancer progression [44]. In addition to cancer-secreted lncRNAs, it has been shown that lncRNAs may be secreted also by other cell types in the TME and may have functional relevance on cancer cells’ behavior. RN7SL1 and HISLA are two relevant lncRNAs, secreted, respectively, by CAFs and TAMs, involved in chemoresistance through different mechanisms. RN7SL1 causes the activation of an anti-viral signaling in the cancer cell, then leading to tumor growth and therapy resistance [45], while exosomal HISLA causes the activation of HIF1A, which favors glycolysis and chemoresistance in the targeted cancer cells [46]. Recently, the presence of lncRNA MALAT1 has also been detected in exosomes from various cancer cell lines, included TNBC cells, and it was shown that exosomal MALAT1 exerts an autocrine pro-proliferative role on MDA-MB-231 breast cancer cells [47]. The function of lncRNA MALAT1, upregulated and oncogenic in a variety of malignancies, included TNBC [48,49,50], has been extensively studied in cancer cells; however, its role in the cross-talk between different cell types in the TME still has to be revealed, and the identification of exosomal MALAT1 now opens up an extremely fascinating and unexplored area of research.

## 3. Circulating Non-Coding RNAs as Biomarkers in TNBC

As mentioned above, EVs have been highlighted as important mediators in the communication among tumor cells as well as between tumor and stromal cells. LncRNAs, miRNAs and circRNAs are encapsulated in EVs and then transferred to proximal and distal recipient cells, inducing responses in the TME both in early and late stages of the tumor progression [51,52]. Moreover, non-coding RNAs may be present in the bloodstream as EV-free molecules, frequently included in ribonucleoprotein complexes or in complexes with lipids or lipoproteins as triglycerols, cholesterol and fat-soluble vitamins [53,54]. Circulating RNAs can also survive in extreme pH conditions, as those present in extracellular environment, and this enables their detection in a variety of biological fluids, such as blood, urine, tears, cerebrospinal fluid, saliva, and semen [52]. These features have made it possible to explore and develop liquid biopsy approaches for diagnosis purposes and to define the prognosis and therapeutic decisions in many tumor types, including TNBC.

The possibility of using new and powerful technologies, such as single-cell RNA sequencing and mass cytometry, enabled shedding light on ncRNA-related dynamic changes of TME components during TNBC transformation and malignant progression [23]. The identification and the full understanding of circulating ncRNA roles in the TME could strongly help in assessing the risk of relapse and metastasis, the response to treatment and in developing new molecular targeted therapies to improve the survival in TNBC.

### 3.1. Circulating microRNAs in TNBC

Many studies that have used RNA sequencing methodologies have highlighted specific miRNA panels implicated at different levels in tumor progression and specifically associated to molecular and histological breast cancer subtypes, redefining the BC hallmarks [55].

The best-known oncogenic miRNA, transversal between the various types of cancer, is miR-21, expressed at high levels in BC II/III stages, HER2 positive and TNBC. Functionally, miR-21 high levels are anti-apoptotic and contribute to the proliferation of cancer cells by inducing the PI3K/Akt pathway, being *PTEN* and *PDCD4* two major targets of this miRNA [56,57,58]. Furthermore, high levels of miR-21 are associated with poor prognosis in patients with TNBC [59]. miR-21 is an independent prognostic factor for overall survival (OS) and disease-free survival (DFS) and predicts the presence of lymph node metastases in TNBC [60,61]. miR-21 has been identified as a circulating miRNA (ci-miRNA) in the serum of breast cancer patients in several studies and a meta-analysis evidenced that increased circulating miR-21 is a potential biomarker for breast cancer [62,63].

One of the first research studies in which serum samples from primary ductal TNBC patients were analyzed for ci-miRNA levels is attributable to a collaboration study of the Børresen-Dale and Santarpia groups [64]. The aim of this research was to identify ci-miRNAs able to predict clinical outcome in TNBC. Through genome-wide serum miRNA screening, Sahlberg and colleagues identified a very robust ci-miRNA signature (miR-18b, miR-103, miR-107, and miR-652) that was able to predict tumor relapse and overall survival. Multivariate Cox regression analysis showed that this four-miRNA signature was an independent prognostic classifier of patients with TNBC [64].

Subsequently, many groups have focused on the research and validation of ci-miRNAs to stratify BC subtypes, to correlate them with the response to therapy and to evaluate their prognostic value. Another ci-miRNA identified in TNBC is miR-720. Increased serum levels of miR-720 were found in TNBC patients highly expressing ADAM8, a protein correlated with invasive and metastatic features in TNBC [65]. Contrarily to the above-mentioned miRNAs, miR-34 and miR-940 were found down-regulated in TNBC vs. healthy controls and miR-940 was also identified as predictor of worse prognosis [66,67].

As TNBCs do not express hormone receptors and HER2, the therapy options for these patients are restricted to neoadjuvant chemotherapy (NAC), radiotherapy, adjuvant chemotherapy and surgery [68]. The aim of NAC is the regression and containment of breast cancer and axillary lymph nodes. Achieving this response at the time of surgery is an important surrogate marker for patient survival prognosis. In this context, the expression of ci-miRNAs as markers of the efficacy of neoadjuvant therapy is an ambitious goal. To date, few studies have connected serum ncRNA levels to the response to NAC. Liu et al. reported that a decrease of serum miR-21 level after NAC plus trastuzumab treatment associates with better outcome (OS and DFS) in patients with HER2-positive breast cancer [69]. Furthermore, another study by Gu et al. highlighted that a low level of miR-451 in serum associates to NAC in a cohort including all breast cancer subtypes [70]. Accordingly, an association between elevated serum levels of miR-451 and improved clinical and pathological response to NAC of locally advanced BC has been identified [71].

Very recently Ritter and colleagues analyzed intra- and extra-cellular breast cancer-related miRNAs in TNBC cells lines treated with various chemotherapeutic agents (carboplatin, paclitaxel, gemcitabine and epirubicin), highlighting that miRNA expression is strongly affected by these treatments. Analysis of the same miRNA panel in serum samples from a small group of TNBC patients (*n* = 8) before and during NAC (pilot study) interestingly evidenced an upregulation of miR-17, miR-19b and miR-30b in those patients who did not achieve a clinical complete response (cCR) to NAC [72]. Despite the small samples size and the preliminary nature of this study, these results lay an optimistic basis for further investigations on the use of miRNAs in liquid biopsy to determine patient response to treatments.

Diverse miRNAs have been recently shown to act in a synergistic way with chemotherapy drugs to decrease cancer proliferation [73]. Thus, the combined use of therapies and selective inhibitors with miRNAs may represent an opportunity to decrease drug resistance in TNBC [74,75].

Very recently, Qattan and colleagues identified through an integrated network analysis a pool of ci-miRNAs that were differentially regulated in TNBC versus normal breast and luminal breast cancer. Furthermore, they highlighted the clinical relationship between some specific ci-miRNAs, chemoresistance pathways, and clinical outcomes [76]. Briefly, miR-19a/b-3p, miR-25-3p, miR-22-3p, miR-210-3p, miR-93-5p, and miR-199a-3p, present at high levels in the blood of TNBC patients, regulate several cancer-related pathways, as PI3K/Akt/mTOR, HIF-1, TNF, FoxO, Wnt, and JAK/STAT, PD-1/PD-L1 and EGFR tyrosine kinase inhibitor resistance (TKIs) [76]. Of note, a significant association of miR-93, miR-210, miR-19a, and miR-19b upregulation with overall survival in these TNBC patients was shown.

ci-miRNAs could also act as key regulators in immune surveillance and immune escape as well as players in metastasis of breast cancer cells. Thomopoulou and colleagues have recently published that the differential expression of plasma miR-10b, miR-19a, miR-20a, miR-126 and miR-155 is able to regulate the immune response during breast cancer progression [77]. They obtained plasma samples from early and metastatic breast cancer patients before adjuvant or first-line chemotherapy, respectively. Low miR-10b and miR-155 levels are associated with shorter disease-free survival, and, in the subgroup of TNBC patients, low miR-155 expression independently predicted short DFS [77].

The results of these recent papers highlight circulating miRNAs as powerful indicators of drug resistance pathways and their potential usefulness as targets for overcoming drug resistance in TNBC.

### 3.2. Circulating lncRNAs in TNBC

LncRNAs are 200 nucleotides or more in length and usually are non-protein-coding transcripts [78]. Numerous sequencing results from cancer patients have now made it clear that lncRNAs expression is deregulated in various types of tumors, included breast cancer [79]. Several lncRNAs are involved in aberrant cell proliferation, apoptosis, invasion, and angiogenesis in cancers [80]. In TNBC, various groups have identified a substantial number of deregulated lncRNAs that could play important roles in the process of tumorigenesis and metastasis. The potential value of these lncRNAs could provide clues for the diagnosis and treatments of TNBC (reviewed in [81]). Although most studies have brought miRNAs to the fore as potential biomarkers, several recent studies have also highlighted the importance of lncRNA analysis as a non-invasive approach for screening and managing BC molecular subtypes.

HOTAIR is a lncRNA that induces migration and invasion of TNBC cell lines and was the first lncRNA to act as a marker of metastasis, also in breast cancer [82,83,84]. Its oncogenic activity is carried out through different mechanisms, such as the regulation of chromatin conformation, by acting as molecular scaffold, or the direct sponging of miRNAs to release expression of target mRNAs [85]. Interestingly, the analysis of tissue samples by in situ hybridization evidenced that HOTAIR expression was associated with lymph node metastasis in a cohort of TNBC samples; of note, HOTAIR was also strongly associated to androgen receptor expression, highlighting its relevance in the LAR subtype of TNBC [86]. Overexpression of serum exosomal or serum circulating HOTAIR has been also repeatedly evidenced and found correlated with poor survival and poor response to chemotherapy in breast cancer patients [87,88,89]. However, liquid biopsy studies did not evidence specific associations between HOTAIR levels and receptors status, suggesting that circulating HOTAIR might represent a powerful liquid biopsy biomarker for breast cancer independently from the subtype [87].

Comparison of lncRNA profiles in plasma samples from TNBC and non-TNBC recently revealed additional lncRNAs that could be used as diagnostic biomarkers in TNBC. Specifically, authors highlighted that antisense noncoding RNA in the INK4 locus (*ANRIL*), hypoxia inducible factor 1alpha antisense RNA-2 (*HIF1A-AS2*), and urothelial carcinoma-associated 1 (*UCA1*) were markedly up-regulated in the plasma of patients with TNBC, compared with patients with non-TNBC. The use of these three lncRNAs as signature showed excellent diagnostic performance [90].

Recently, the expression analysis performed in the serum samples from 72 TNBC, 105 non-TNBC, 60 benign breast disease patients and 86 healthy subjects evidenced that higher lncRNA TINCR levels are detectable in BC patients, especially in TNBC, compared to subjects without cancer [91]. High circulating TINCR was significantly correlated with clinicopathological features and with poor OS in TNBC. High TINCR expression distinguished the subgroup of TNBC patients with cancer relapse. Interestingly, it has been subsequently demonstrated that TINCR and miR-761 act along a functional axis in early TNBC, promoting cell migration, invasion and EMT [92]. Furthermore, the authors showed that luteolin (LU), a natural compound with anti-TNBC activity, was able to repress the TINCR/miR-761 axis impinging on their metastatic potential [92]. These results indicate that TINCR/miR-761 targeting could represent a potential therapeutic approach for TNBC.

An epigenome-wide association study (EWAS) conducted on a large cohort of TNBC patients to identify circulating biomarkers showed that LINC00299/ID2 (RNA 299) had a higher methylation in TNBC patients compared with controls [93]. The fact that hypermethylation of LINC00299 in peripheral blood could represent a useful circulating biomarker for TNBC was also supported by Manoochehri and colleagues who analyzed an additional prospective cohort of patients [94]. Interestingly, they found a significant association between methylated LINC00299 levels and TNBC subgroup in young age patients (age 26–52 showing *p* = 0.0025 and age 22–46 showing *p* = 0.001, respectively). These results suggest a potential role of hypermethylated LINC00299 as diagnostic biomarker in young women with TNBC [94].

LncRNA X-inactive specific transcript (XIST) is an important regulator for X inactivation in mammals [95]. Recently, it was observed that XIST plays critical roles in tumor growth and gene expression control, also thanks to its ability to act as a miRNA sponge [96]. Moreover, XIST has been shown to act as a prognostic factor in diverse cancer types [97]. A very recent study explored the clinical value of exosomal XIST secreted in the serum by tumor cells to predict recurrence in patients with TNBC [98]. Of note, TNBC tissues and blood serum samples from relapsing patients showed higher XIST and exo-XIST expression level compared to non-recurrent patients. According to the rationale of the study, the authors found that exo-XIST was expressed at low level after resection of the primary breast tumors and at high level at the time of recurrence. Importantly, serum exo-XIST expression was associated with poor overall survival of TNBC patients [98]. This interesting study correlated the serum exo-XIST to the diagnosis and prognosis of TNBC patients and showed that serum exo-XIST may be a valid biomarker to predict the recurrence status.

### 3.3. Circulating Circular RNAs in TNBC

With the rapid development of RNA-seq methodologies, the related bioinformatics analyses and the containment of sequencing costs, more and more information has been gathered on circular RNAs (circRNAs). These data have highlighted circRNAs as very stable molecules expressed in all eukaryotes, highly circulating in TME and fluids, and very versatile and specific, in terms of cell type, tissue or developmental stage, in the regulation of numerous physiological pathways [99,100,101]. Thanks to these characteristics, circRNAs have recently been judged to be ideal candidates as biomarkers of diagnosis and prognosis, particularly in liquid biopsies [20,102,103]. Since circRNAs were the last molecules of the ncRNA family to emerge as functionally relevant, few of these have been so far analyzed in serum or plasma, and even less in other biological fluids [104].

A systematic review and meta-analysis has been recently carried out to evaluate the value of circRNAs in the diagnosis of breast cancer. This study considered all the published articles reporting about the detection of circRNA expression levels in serum, plasma, or tissue before 31 December 2020 [105]. In this study, circRNAs exhibited a high diagnostic power for breast cancer, with two circRNAs, including circ_0001073 and circTADA2A-E5/E6, showing the highest diagnostic values, with AUC value of 0.990 and 0.937, respectively [105]. These results are prompting scientists to implement the search for circRNAs in the fluids of breast cancer patients and to assess their potential use as biomarkers in the various subtypes of BC.

A few studies have recently explored the expression level of hsa_circ_0000615 (circZNF609), hsa_circ_0104824, hsa_circ_0069094, hsa_circ_0079876, hsa_circ_0017650, and hsa_circ_0017526 in the peripheral blood of breast cancer patients and assessed their diagnostic value [106,107,108]. In a cohort of 57 BC patients, the plasma level of hsa_circ_0001785 was related to the histological grade, TNM stage, and distant metastasis of breast cancer, with high diagnostic value (AUC = 0.784) [109]. Wang et al. reported the significant overexpression of hsa_circ_0020707, hsa_circ_0064923, hsa_circ_0104852, hsa_circ_0087064, and hsa_circ_0009634 in the serum from patients with breast cancer and positive correlations with carcinogenesis and progression [110].

In addition to the assessment of the expression level and clinical relevance of circulating circRNA, a few studies have been recently published that reported the evaluation of specific circRNA in serum samples along with their functional characterization at the intracellular level. Identification of circRNA function adds relevant information if these circulating molecules are to be used as targets for cancer diagnosis and treatment.

A few examples of circRNA present in the blood of BC patients and with known functional roles in TNBC cells are described below and in Figure 4.

Jia et al., found higher levels of circKIF4A (hsa_circ_0007255) in the tissue and serum samples from BC patients compared to healthy controls and adjacent normal tissues, respectively. Authors identified in BC cells an oncogenic role for circ_0007255, which is responsible for the inhibition of miR-335-5p and the consequent release of expression of its target SIX2, with functional impact on oxygen consumption, colony formation, and cell motility in BC cells [111]. An oncogenic role of circKIF4A has been reported by Tang et al., who confirmed that circKIF4A is specifically overexpressed in TNBC tissues, where it is associated to poor clinical outcome. In TNBC cells, circKIF4A also favors the expression of KIF4A by sponging miR-375 to impinge finally on the proliferation and migration of TNBC cells [112].

Interestingly, circBCBM1 (hsa_circ_0001944) overexpression in primary BC tissues was associated with shorter brain metastasis-free survival and with brain metastasis promotion in mouse model [113,114]. Accordingly, circBCBM1 was also found upregulated in plasma of BC patients who developed brain metastasis, becoming a putative novel diagnostic and prognostic biomarker and potential therapeutic target for breast cancer brain metastasis [114]. Functionally, circBCBM1 acts by sequestering miR-125a, enabling released expression of BRD4 protein with subsequent up-regulation of MMP9, a crucial player in the metastatic process.

Very recently, the novel circular RNA circHIF1A (circ_0032138) was found overexpressed in breast cancer tissues and associated with TNBC subtype, metastasis, and poor prognosis [115]. By a mechanistic point of view, circHIF1A modulated the expression and translocation of transcription factor Nuclear Factor I B (NFIB), through post-transcriptional and post-translational modifications, leading to the activation of the Akt/STAT3 signaling pathway and inhibition of p21. These activities are related to increased proliferation and invasion in TNBC cell models [115]. Interestingly, the authors assessed the presence of circHIF1A into exosomes and found it up-regulated in the plasma of breast cancer patients [115]. In support of circHIF1A as a biomarker and target molecule for breast cancer therapy, Zhan and colleagues identified this molecule in a screening of circRNA present in exosomes from hypoxic CAFs in breast cancer [116]. circHIF1A from hypoxic CAFs-derived exosomes proved to be an important player in conferring stem cell properties to breast cancer cells, by sponging miR-580-5p and consequently upregulating CD44 expression [116].

Additionally, circPSMA1 was found up-regulated in the exosomes from serum of TNBC patients and TNBC cell lines, compared to non-TNBC patients and non-TNBC cell lines. Functionally, intracellular circPSMA1 acts by sponging miR-637, releasing the expression of its target Akt1 [117].

More and more studies are associating altered expression of circRNAs with protumoral and metastatic functions in TNBC [118,119,120]. For example, circWAC was found highly expressed and associated with worse TNBC patient prognosis [121]. It exerted the oncogenic activity by affecting miR-142/WWP1/PI3K signaling and inducing resistance to chemotherapeutic treatment with paclitaxel (PTX) in vitro and in vivo [121]. Novel circPDCD11 (hsa_circ_0019853) was significantly upregulated in TNBC tissues and cells and closely correlated with a poor prognosis, acting as an independent risk factor for TNBC prognosis [122]. Functionally, circPDCD11 was proved to accelerate glucose uptake, lactate production, ATP generation, and the extracellular acidification rate in TNBC cells, enhancing LDHA expression by sponging miR-432-5p [122].

All these circRNAs identified in tumor and metastatic TNBC tissues would be promising biomarkers in liquid biopsy if they were identified also in human fluids, creating a panel of robust biomarkers for the early diagnosis and prognosis and active surveillance of TNBC patients.

## 4. Conclusions and Perspectives

Secreted non-coding RNAs play crucial roles during cancer progression and strongly contribute to remodel the tumor microenvironment and the metastatic niche, to enable the formation of a supporting vasculature, the inhibition of tumor recognition by the immune system and, finally, the spreading of tumor cells and metastatization. The full comprehension of the ncRNA-guided networks at the basis of these events is central for the development of novel effective therapies aimed at disrupting the cross-talk between tumor cells and other cell types in the tumor microenvironment; such therapeutic approaches would strongly prompt the immune system to recognize and eliminate tumor cells. At the same time secreted non-coding RNAs also represent powerful biomarkers to be exploited for diagnostic in liquid biopsy and for therapeutic purposes. A comprehensive understanding of the mechanisms of action of secreted ncRNAs in TNBC represents the future challenge, which will allow the widest use of these molecules both as diagnostic tools and as therapeutic targets.

## Figures and Tables

**Figure 1 ncrna-08-00005-f001:**
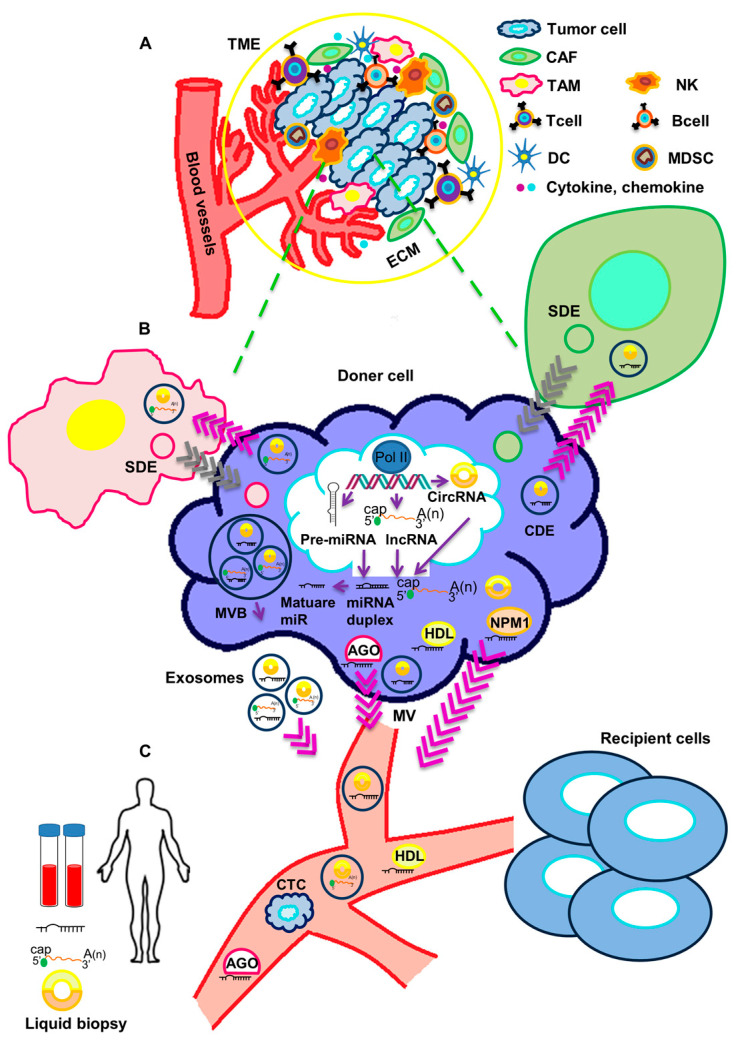
Crosstalk of circulating non-coding RNAs and tumor microenvironment. (**A**) The tumor microenvironment (TME) includes a cellular and an extracellular component. The cellular component consists of tumor cells, immune cells (TAMs, T cells, B cells, NK cells, DCs, and MDSCs) and cancer-associated fibroblasts (CAFs). The extracellular component of TME is composed by ECM proteins and signaling molecules (like cytokines, chemokines, growth factors, hormones, etc.), which are secreted by the cellular component. (**B**) ncRNAs are processed and released in body fluids. Circulating ncRNAs can be released in two ways: (1) as cell-free RNAs, complexed with protein such as AGO, NPM1 or HDL; or (2) in extracellular vesicles (exosomes and microvesicles). Both tumor cells and stromal cells may release exosomes in the TME. Through body fluids, ncRNAs can reach distant sites in the body and act as molecular mediators. (**C**) Circulating ncRNAs are stable and easily detectable and can be used as non-invasive biomarkers in liquid biopsy. TME: tumor microenvironment, TAM: Tumor-associated macrophage, NK: Natural killer cell, DC: Dendritic cell, MDSC: Myeloid-derived suppressor cell, CAF: cancer-associated fibroblast, ECM: extracellular matrix, SDE: Stromal-derived exosome, CDE: Cancer-derived exosome, AGO: Argonaute2, NPM1: Nucleophosmin 1, HDL: High-density lipoprotein, MVB: Multivesicular body, MV: Microvesicle, CTC: Circulating tumor cell.

**Figure 2 ncrna-08-00005-f002:**
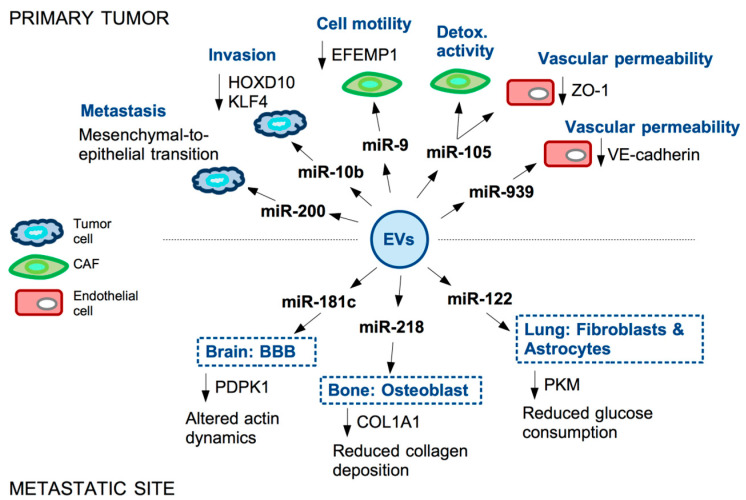
Examples of miRNAs secreted in EVs by TNBC cells and their functional impact in the primary tumor and at metastatic site. microRNAs are secreted by TNBC cells into extracellular vesicles (EVs). These reach recipient cells both in the tumor microenvironment (TME) and, through the bloodstream, in the metastatic niche. miRNAs enclosed in the EVs contribute to tumor progression by affecting motility and metabolism of tumor cells and fibroblasts (CAFs), permeability of the vasculature and stiffness of the TME.

**Figure 3 ncrna-08-00005-f003:**
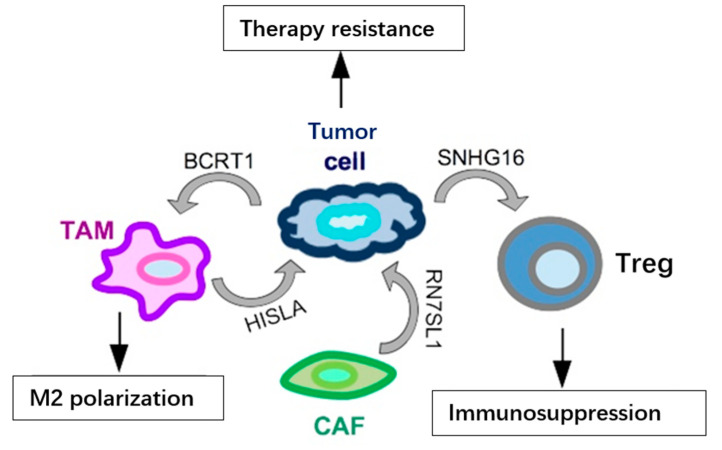
Examples of lncRNAs secreted in exosomes by cells in the TME and their functional output in TNBC. LncRNAs (BCRT1 and SNHG16) may be secreted by cancer cells and impinge on the activity of other cell types in the TME, such as TAMs and Tregs, enhancing their pro-tumoral behavior. At the same time, various cells in the TME may release lncRNAs, such as HISLA and RN7SL1, in exosomes and reach tumor cells to increase their resistance to treatments.

**Figure 4 ncrna-08-00005-f004:**
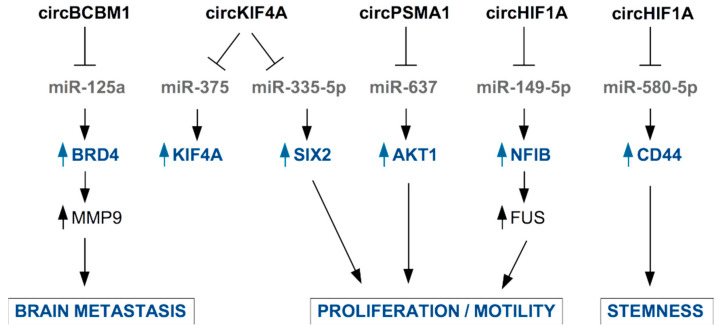
Examples of circRNAs secreted in EVs by TNBC cells or by CAFs and their functional role in TNBC cells. CircRNA are mainly involved in the inhibition of the function of microRNAs through sponging activity. This results in released expression of miRNA’s target mRNAs and enhancement of pro-tumoral properties.

## Data Availability

Not applicable.

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
