# Peer review of "Secreted Non-Coding RNAs: Functional Impact on the Tumor Microenvironment and Clinical Relevance in Triple-Negative Breast Cancer"

_ncrna, 2022, doi:10.3390/ncrna8010005_

Round 1

Reviewer 1 Report

This manuscript reviews the function of secreted circulating non-coding RNAs (ncRNAs) in the tumor microenvironment and their possibilities of clinical use as diagnostic and prognostic non-invasive biomarkers in triple-negative breast cancer (TNBC).

Please cover more lncRNAs such as MALAT1, HULC, HOTAIR etc., in section 3.2, and add a figure. 

The conclusion section with perspective should be added.

On line 32, “anddefined” should be corrected.

On line 242, “ci​-miR-21” should be defined.

Author Response

Following the reviewers' suggestions we have developed in deeper detail the description of secreted lncRNAs. In particular:

  • We have enclosed in paragraph 2 (Functional impact of secreted ncRNAs on surrounding stromal cells and at metastatic sites in TNBC) information relative to secreted lncRNAs which have been reported to have a functional role in the tumor microenvironment in the cross-talk between different cell types; we also included a figure (new Figure 3) depicting these cross-talks.
  • We have enclosed in paragraph 3.2 (Circulating lncRNAs in TNBC) information relative to secreted lncRNAs that have been detected in biofluids in TNBC and may be relevant as biomarkers.

Moreover, we have included the "Conclusion and perspectives" paragraph.

All the changes and new text are marked in blue.

We hope that our review article is now suitable for publication in ncRNA journal.

Reviewer 2 Report

This is a good review, it will be helpful to the community. 

Author Response

(The authors gave the same response as above.)
